# Physico-Chemical and Microbiological Control of the Composting Process of the Organic Fraction of Municipal Solid Waste: A Pilot-Scale Experience

**DOI:** 10.3390/ijerph192315449

**Published:** 2022-11-22

**Authors:** Natividad Miguel, Andrea López, Sindy D. Jojoa-Sierra, Julen Fernández, Jairo Gómez, María P. Ormad

**Affiliations:** 1Water and Environmental Health Research Group, University Institute for Research in Environmental Sciences of Aragon, Department of Chemical Engineering and Environmental Technologies, University of Zaragoza, María de Luna 3, 50018 Zaragoza, Spain; 2Navarra de Infraestructuras Locales S.A. (NILSA) Avda, Barañain 22, 31008 Pamplona, Spain

**Keywords:** compost, OFMSW, microbiological control, composting technologies

## Abstract

The aim of this work was to carry out a pilot experiment to monitor OFMSW (organic fraction of municipal solid waste) composting processes using different types of installations (automatic reactor, aerated static pile and turned pile). To carry out the process, pruning waste was used as structuring material (SM), in a 1:1 and 1:2, *v*:*v*, OFMSW:SM ratio. Monitoring was carried out through the control of physico-chemical and microbiological parameters, such as temperature, pH, humidity, Rottegrade, Solvita tests, the presence of *Salmonella* sp. and *Escherichia coli*, total coliform, and *Enterococcus* sp. concentrations. After carrying out the tests, it can be affirmed that the three types of installations used worked correctly in terms of the monitoring of physico-chemical parameters, giving rise to a compost of sufficient stability and maturity to be applied on agricultural soil. In all cases the bacterial concentrations in the final compost were lower than those detected in the mixture of initial components for its preparation, thus complying with the requirements established in RD 506/2013 and RD 999/2017RD on fertilizer products. However, it cannot be affirmed that one of the three types of installation used produces a greater bacterial inactivation than the others. When composting with different types of facilities, it is of interest to optimize the irrigation and aeration system in order to have a better control of the process and to study the possible temperature gradients in the piles to ensure good sanitization without the risk of bacterial proliferation a posteriori. Finally, the different initial mixtures of OFMSW and SM used in this study did not have a significant influence on the functioning of the composting process or on the microbiological quality during the process. The irrigation water can provide a bacterial contribution that can lead to increases in concentration during the composting process. This study is part of the Life-NADAPTA project (LIFE16 IPC/ES/000001), an integrated strategy for adaptation to Climate Change in Navarra, where NILSA participates in water action and collaborates in agricultural action, which includes among its objectives the development of new soil amendments from different organic waste.

## 1. Introduction

Municipal waste is made up of different components, with its organic fraction being predominant. This fraction represents a percentage by weight of 44–49% of waste [1] and is an important element in the objectives established in Spanish Law 22/2011 on waste and contaminated soils, which aims to promote its separate collection, its treatment for the production of compost, and its subsequent use in the agricultural sector [2]. In this way, it promotes the reduction of biodegradable municipal waste destined for landfill, another of the objectives of the aforementioned law, and contributes to the Thematic Strategy on the Sustainable Use of Natural Resources and the European Climate Change Program.

In Spain, around 18 million tons of municipal waste was collected in 2018, 17% of which was composted. In the same year, one million tons of biodegradable waste was collected separately, 65% of which was composted [1].

The organic fraction of waste can be treated by biological processes, both aerobic (composting) and anaerobic, in order to hygienize it and stabilize the organic matter. In the case of composting, woody plant residues (pruning) are fundamental since their function is that of structuring materials (i.e., to favor the appropriate carbon/nitrogen ratio and to provide structure to the mixture to facilitate the presence of oxygen in the process) [3].

The composting process can be carried out in different types of facilities: open systems such as windrows or piles, either aerated static or turned; and closed systems such as reactors or composters [4]. In all cases, appropriate pH, temperature, and humidity conditions are necessary for the composting process to take place through the mesophilic, thermophilic, cooling, and maturation phases characteristic of this type of process [5].

The resulting material, compost, is a stable material that can be used as an agricultural amendment as it brings many benefits to soils: it provides organic matter and biofertility, improves soil structure and water retention, decreases erosion, etc. [1].

The use of compost in agricultural soils is regulated in Spain by Royal Decree 506/2013 on fertilizer products [6], amended by Royal Decree 999/2017 [7]. The law establishes maximum admissible concentrations of microorganisms and heavy metals. In terms of microorganisms, it establishes that Salmonella must be absent in 25 g of compost and the concentration of Escherichia coli must be less than 1000 (most probable number) per gram of compost. There are other regulations in European countries that include other microbiological parameters, although Salmonella and Escherichia coli are the most common [8]. With regard to heavy metals, maximum concentrations of cadmium, cobalt, nickel, lead, zinc, mercury, and chromium have been established. However, the current legislation does not consider other organic, inorganic and microbiological contaminants that can be found in the initial waste and that could have different negative effects on the environment and on human health [9,10,11,12].

In the Autonomous Region of Navarra, where this study was carried out, the “Waste Plan of Navarra 2017–2027” was drawn up, in accordance with current legislation on waste, whose objectives include “Moving towards high quality selective collection, maximizing waste recovery and recycling and guaranteeing the co-responsibility of the waste generator (citizens, companies, etc.) and taking into account the principle of gender equality, through the usual management routes and through emerging processing routes, especially for domestic bio-waste and industrial waste”.

Navarra de Infraestructuras Locales S.A. (NILSA), a company which specializes in sanitation and water treatment and manages the Waste Consortium of the Autonomous Region of Navarra, in collaboration with the Water and Environmental Health group, a reference research group recognized by the Government of Aragon and belonging to the University Institute for Research in Environmental Sciences of Aragon, is developing a project (to which this study belongs) to promote the correct management of bio-waste in Navarra, encouraging its recycling through composting processes, from a safe environmental and health perspective. Its objective is the application of composting processes in the treatment and recovery of organic waste, with the aim of promoting its use as an agricultural fertilizer, while minimizing the associated environmental and health risks. The project also aims to establish guidelines for action that facilitate the adoption of good agricultural practices for the application of the different fertilizers obtained and that encourage coordination of the strategy for the management of organic waste by the public sector with the needs of the agricultural sector.

Specifically, the aim of this work is to study the influence on the physico-chemical and microbiological characteristics of the compost obtained by the type of installation used in the composting process at pilot scale (automatic reactor, aerated pile, and turned pile) and of the characteristics and mixing ratio of the initial waste used (organic fraction of municipal solid waste and pruning material as a structuring material).

## 2. Materials and Methods

### 2.1. Municipal Waste, Structural Material and Irrigation Water

The waste used in this study came from a waste treatment plant in Navarra, which processes the organic fraction of municipal solid waste (OFMSW). Pruning waste was used as a structuring material (SM) for composting and the water used to provide moisture for the process was taken from the water level.

### 2.2. Facilities Used for Composting

OFMSW composting was carried out in three types of pilot facility with the operating conditions described below:

#### 2.2.1. Automatic Reactor

The reactor has two compartments of approximately 125 L capacity with polyurethane insulation walls and a longitudinal axis connected to the aeration system. Both aeration and turn-over can be carried out automatically by means of a microprocessor. Water was added manually. The temperature was controlled automatically by means of PT100 sensors. A picture of the automatic reactor can be seen in Figure 1.

#### 2.2.2. Aerated Static Piles

Two aerated static piles of 4–6 m^3^ capacity were used. Air was supplied through 3 pipes installed in the floor of the piles and water through a drip system until the piles reached a humidity of 40%. Covers were placed over the piles to help maintain humidity. A PT100 sensor was used to automatically monitor the temperature of the piles and the ambient temperature. Figure 2 shows images of the assembly process of the aerated piles.

#### 2.2.3. Turned Piles

The turned piles had a capacity of 4–6 m^3^. Aeration was carried out manually by mechanical turning of the piles with a shovel and water was supplied manually through a water hose until the pile reached a humidity of 40%. The piles were covered with layers to help maintain humidity. A PT100 sensor was used to automatically monitor the temperature of the piles and the ambient temperature. Figure 3 shows images of turned piles.

#### 2.2.4. Operational Conditions

OFMSW composting was carried out in the facilities described above, with different initial ratios between the OFMSW and SM (approximate ratio of 1:2 and 1:1 *v*/*v*), in function of the available pruning material and the relation C/N of the mixture. This is summarized in Table 1. Aeration and water addition was carried out as needed throughout the process.

### 2.3. Sampling

Sampling of compost and the initial materials for composting (OFMSW and SM) was carried out according to the ‘Test Methods for the Examination of Composting and Compost’ [13]. Different sample portions were taken at random and mixed together. The mixture was then homogenized and the quartering method was applied in order to obtain a final representative sample of 500 g for further analysis. Water sampling was performed according to the standard method ISO 5667-3:2018 [14].

### 2.4. Analytical Methodology

The solid samples were pre-treated to analyze the microbiological and physico-chemical parameters. This pre-treatment was based on that described by Carter (1993) [15]. For the determination of the microbiological parameters, 10 g of solid sample was taken and 90 mL of distilled water was added. The mixture was then triturated for 5 min and decanted to separate the solid fraction from the liquid. For the determination of the physico-chemical parameters, 20 g of solid sample was taken and 100 mL of distilled water was added. The mixture was shaken for 2 h, centrifuged at 12,000 rpm for 20 min and the sample filtered through a 1.5 µm Whatman filter. The liquid samples resulting from this pre-treatment and the water samples used in the study were analyzed according to the standard methodology described below.

#### 2.4.1. Physico-Chemical Parameters

The samples analyzed in this part of the study were: the initial OFMSW, the initial SM, the mixtures of initial OFMSW and SM at different ratios (1:1 and 1:2 *v*/*v*), and the compost samples throughout the four months of the composting process. All of these samples were analyzed in the three facilities used in this study (automatic reactor, aerated static pile, and turned pile).

Initially, a characterization of the OFMSW and the structuring material was carried out by analyzing humidity, total solids, volatile fraction, density, pH, organic matter, nitrogen, phosphorus, and heavy metals. In order to control the composting processes, an automatic temperature control was carried out once the process was started, as well as a pH and humidity control. All of the physico-chemical parameters analyzed, the equipment used, and the standard methodology of analysis are shown in Table 2.

Finally, a study of the maturity and stability of the compost obtained was carried out using two techniques: the determination of the degree of Rottegrade (UNE-EN 16087-2:2012) [18], which establishes a classification of compost maturity based on the maximum temperature reached in 10 days, with a constant ambient temperature of 20 °C and a sample with a humidity of 40%; and using a Solvita^®^ test, which allows the stability of the compost to be evaluated through qualitative measurements of CO_2_ and ammoniacal nitrogen using a colorimetric technique. These types of tests are widely used to assess the stability of compost from all types of waste [19,20], classifying them according to their degree of maturity (degree of maturity referring to resistance to decomposition, and absence of ammonia, organic acids, and phytotoxic components).

#### 2.4.2. Microbiological Parameters

The samples analyzed in this part of the study were the same as those used in the analysis of the physico-chemical parameters and, in addition, samples of the water used to provide humidity during the composting process.

The initial microbiological characterization of the materials used for composting, OFMSW and SM, was carried out through the analysis of six bacteria: total coliforms, *Escherichia coli*, *Enterococcus* sp., *Clostridium perfringens*, total mesophiles and *Salmonella* sp. Among these bacteria, those selected for monitoring microbiological parameters during the composting process and in the irrigation water were *Salmonella* sp. and *Escherichia coli*, as they are the microorganisms for which maximum concentrations in the compost are established by current legislation, and total coliforms and *Enterococcus* sp., due to their staining characteristics (gram negative and gram positive, respectively) and since they are parameters commonly used in the microbiological control of environmental matrices.

The microbiological parameters analyzed in the samples, the culture media used, and the standard methods of analysis are shown in Table 3.

To reliably determine the concentration of bacteria in the samples, serial dilutions were performed. These dilutions were carried out by dissolving 1 mL of sample in 9 mL of 0.9% NaCl. 

All samples were analyzed using the plate count method. After surface seeding or using the membrane filtration method, the samples were subjected to the appropriate incubation period for each bacterium (time and temperature), resulting in plates with colored colonies that could be counted as colony forming units (CFU). The microbiological concentration of solid samples is expressed as CFU per gram of dry matter (measured as total solids) and the microbiological concentration of water samples is expressed as CFU per 100 mL.

To determine the fertilizing capacity of the compost obtained, the germination index (GI), representative of the phytotoxicity of the compost [8], was analyzed, following the method developed by Zucconi et al. (1981) [25]. The GI of *Lepidium sativum* L. seeds was analyzed on three different bases: distilled water, unamended soil, and commercial peat.

## 3. Results and Discussion

### 3.1. Initial Characteristics of OFMSW, SM and Irrigation Water

The initial physico-chemical properties of OFMSW and SM used in the composting are shown in Table 4.

As can be seen in terms of the physicochemical properties of OFMSW and SM, the main differences lie in humidity, which is higher in OFMSW, total solids, which are higher in SM, and nitrogen and phosphorus, which have higher values in OFMSW. The rest of the physico-chemical parameters present similar values in both components. As for heavy metals, although the values detected in OFMSW and SM are different, at no time are the maximum concentrations established in RD 506/2013 (modified by RD 999/2017) for heavy metals in compost exceeded. The two initial mixtures tested in this study result in C/N ratios between 5.2 and 5.5.

Table 5 shows the bacterial concentrations of the bacteria analyzed in the OFMSW and SM used for composting prior to mixing.

As shown in Table 5, in general OFMSW has higher bacterial concentrations for all the bacteria analyzed than SM, this being especially notable for *Escherichia coli* and *Enterococcus* sp., with the exception of total coliforms, whose concentration is similar in both components. In addition, *Salmonella* sp. was not detected in either of them.

With regard to the microbiological analysis of the irrigation water, used to provide humidity during the composting process, concentrations of 6.2 × 10^2^ to 3.1 × 10^5^ CFU 100 mL^−1^ of total coliforms, 5.0 × 10^1^ to 1.4 × 10^5^ CFU 100 mL^−1^ of *Escherichia coli* and 5.0 × 10^1^ to 1.1 × 10^4^ CFU 100 mL^−1^ of *Enterococcus* sp. were found, concentrations quite variable depending on the time of analysis during the process.

### 3.2. Evolution of Compost Characteristics during Composting Process

#### 3.2.1. Physico-Chemical Properties

The control of the composting processes was carried out by monitoring temperature, pH and humidity. The evolution of these parameters in the three pilot scale composting facilities (automatic reactor, aerated static pile and turned pile) and with the two initial ratios between OFMSW and SM (OFMSW:SM of 1:1 and 1:2 *v*/*v*) is described below.

(a)Automatic reactor

Figure 4 shows the evolution of temperature during the composting process using the automatic reactor as well as the daily average ambient temperature and Figure 5 shows the evolution of humidity and pH, both for the two initial mixtures of OFMSW and SM.

As can be seen in Figure 4, the temperature increases rapidly during the first two weeks, being generally above 45 °C, the period in which aeration in the automatic reactor is more frequent. Subsequently, the temperature tends to stabilize at around 30 °C for the rest of the process. As far as the temperature evolution is concerned, no major differences are observed between the initial mixtures with the ratio OFMSW:SM = 1:1 or 1:2.

Figure 5 shows a decrease in pH, from initial values above 8.0 to final values around 7.5. It is also observed that the humidity is in a range of 30 to 60% during practically the whole process, it being necessary to add water after 10 days for both initial mixtures and on the 24 day of the process for the initial mixture OFMSW:SM = 1:1.

In general, slightly higher humidity values are observed for the initial 1:2 mixture during the process, probably due to the fact that although SM has a much lower humidity than OFMSW initially (see Table 4), a higher amount of SM can provide greater water holding capacity after additions. In both cases, the humidity stabilizes at around 40% at the end of the composting process.

(b)Aerated static pile

Figure 6 shows the evolution of temperature during the composting process using the aerated static pile as well as the daily average ambient temperature and Figure 7 shows the evolution of humidity and pH, both for the two initial mixtures of OFMSW and SM.

As can be seen in Figure 6, the temperature of both piles increases in the first 24 h of the process and remains above 50 °C for the first two weeks. Subsequently, a progressive decrease in temperature is observed until it stabilizes at around 30 °C without significant differences between the two initial mixtures of OFMSW and SM, as was the case in the automatic reactor.

Figure 7 shows that in both piles the pH is between 7.0 and 8.0 during the whole process, obtaining final values around 7.5 for both initial mixtures. In this case, humidity in general is low during the whole composting process (20–40%), despite the water additions made at different times of the process (mainly during the first five weeks, see Figure 6). This may be due to the fact that when using static piles for composting, the absorption in the pile may not be homogeneous due to the formation of channels and preferential pathways. It should be noted that, although the humidity evolution of the piles with the two initial proportions of OFMSW and SM is very similar during almost the first half of the process, the humidity of the compost obtained for the initial mixture OFMSW:SM = 1:2 is very low and lower than that obtained with the other mixture. This fact shows the need to optimize the irrigation system in this pile.

(c)Turned pile

Figure 8 shows the evolution of the temperature during the composting process using the turned piles, as well as the daily average ambient temperature, and Figure 9 shows the evolution of humidity and pH, both for the two initial mixtures of OFMSW and SM.

As shown in Figure 8, the temperature increases in the first days of the process reaching values above 65 °C but showing constant temperature rises and falls during the first four weeks. This period coincides with the addition of water and the turning of the pile which, although carried out throughout the whole period, are more frequent in the first weeks. In general, the pile with the initial OFMSW:SM = 1:1 ratio shows a slightly higher temperature throughout the process. At the end of the process, as with the two previous installations, the temperature stabilizes at around 30 °C for both piles.

Figure 9 shows that both piles have a pH in the range of 7.5–8.0 during the whole composting process. The final pH using the turned pile is 7.7–7.8, slightly higher than that obtained with the automatic reactor and the aerated static pile.

The humidity values are between 20 and 50% during the whole process in both piles, and low values are observed in spite of the multiple additions of water during the process (see Figure 8). The greatest differences between the two piles are observed from the seventh week onwards, when the humidity of the pile with the initial OFMSW:SM = 1:2 ratio is notably higher than that of the 1:1 ratio, although at the end of the process, compost with the same humidity of around 30% is obtained.

Comparing the results obtained in terms of the monitoring of the physico-chemical parameters of the composting process using the three types of facilities, it is observed that the highest temperatures are reached in the turned pile, followed by the aerated static pile and the automatic reactor, although in no case is the temperature corresponding to “microbial suicide” [26] exceeded, at which the biological composting process itself would be inhibited. In all cases the highest temperatures are reached in the first days of the composting process, complying with the proposal for a Regulation of the European Parliament and of the Council laying down provisions relating to the placing on the market of fertiliser products bearing the CE marking. The final temperature of the compost obtained (30 °C) is similar in all the facilities, irrespective of the initial ratio between OFMSW and SM. Similar evolutions of the pH are observed in the three facilities. Finally, the automatic reactor maintains a higher and more suitable humidity during the process than the other two facilities, given that the initial volume of waste treated is considerably lower in the automatic reactor (see Table 1), which means that the process is more easily controlled using this facility. At the end of the process, with all three systems compost with 30–40% humidity is obtained for both initial mixtures of OFMSW and SM (with the exception of the 1:2 mixture with the aerated static pile).

Finally, the Rottegrade and the Solvita^®^ test indicate that the compost obtained in the three types of facility and with different initial mixtures of OFMSW and SM is highly stabilized and sufficiently mature at the end of the process (grade V according to Rottegrade and maturity index 6 according to Solvita^®^). Moreover, the C/N ratio of the final compost obtained from the three facilities has values of 12.5–13.5, complying with the requirements established in the current legislation.

#### 3.2.2. Microbiological Characteristics

Figure 10 shows the evolution of the bacterial concentration (total coliforms, *Escherichia coli* and *Enterococcus* sp.) in the compost during the composting process using the three types of installation (automatic reactor, aerated static pile and turned pile) with both OFMSW/SM ratios. *Salmonella* sp. was not detected at any time in any of the facilities used.

As can be seen, starting from the same initial bacterial concentrations (approximately 10^8^ CFU g^−1^) with the three types of pilot composting facilities, the concentrations are lower at the end of the process, complying with the requirements established in RD 506/2013 on fertilizer products (modified by RD 999/2017). However, the overall bacterial reductions are different depending on the facility used: in the case of the automatic reactor, these concentrations are reduced to 10^2^–10^3^ CFU g^−1^ for all bacteria; using the aerated static pile, the final concentrations vary between 10^3^ CFU g^−1^ for *Escherichia coli* and 10^6^ CFU g^−1^ for total coliforms; and using the turned pile, the final concentrations are between 10^1^ CFU g^−1^ for *Escherichia coli* and 10^5^ CFU g^−1^ for total coliforms, with the concentration of *Enterococcus* sp. being around 10^3^–10^4^ CFU g^−1^ in both piles.

The results for the automatic reactor show a progressive decrease in the concentrations of the three bacteria analyzed, being more pronounced during the first days, probably due to the fact that the first days of the process is when there is a greater increase in temperature in the reactor (see Figure 4). Fifteen days after starting the composting process using the automatic reactor, there is a bacterial reduction of three to four logarithmic units, reaching concentrations of 10^2^–10^3^ CFU g^−1^ of the three bacteria at the end of the process, which means a total reduction of five to six logarithmic units in the concentration of the bacteria. The behavior of the bacteria is similar throughout the process and no significant differences in bacterial evolution are found in the cases of composting with an initial mixture of OFMSW and SM of 1:1 (*v*/*v*) or 1:2 (*v*/*v*).

The aerated static pile results show different behavior depending on the bacteria. The concentration of total coliforms remains practically constant throughout the process; only variations of approximately 1 logarithmic unit are observed. The concentration of *Escherichia coli* decreases progressively throughout the process, with a reduction in concentration of about 5 logarithmic units at the end of the process with respect to the initial concentration of the mixture, this being the bacterium whose concentration is most reduced. In the case of *Enterococcus* sp., concentrations of the order of 10^4^ CFU g^−1^ are reached at the end of the process, which means a reduction of four logarithmic units. Some increases in concentration can be attributed to the bacterial contribution through the irrigation water, since during the process maximum concentrations of 10^4^–10^5^ CFU 100 mL^−1^ are detected for the three bacteria. In addition, if during the thermophilic phase there has not been a homogeneous temperature throughout the pile, bacteria may remain in the compost and proliferate over time. Again, as with the automatic reactor, no significant differences in bacterial evolution are found in the cases of composting with an initial mixture of OFMSW and SM of 1:1 (*v*/*v*) or 1:2 (*v*/*v*).

Finally, the results obtained using the turned pile show a similar behavior as in the case of using the aerated static pile, although the final bacterial concentrations are lower in all cases. The concentrations of total coliforms and *Escherichia coli* decrease, especially at the beginning of the composting process, coinciding with the period in which a higher temperature increase is observed in the piles (see Figure 6 and Figure 8). Specifically, for the turned pile, the decrease in the concentration of these two bacteria is three to four logarithmic units in two weeks, after which a small variation in the concentration of total coliforms is observed and a very pronounced decrease in the concentration of *Escherichia coli*, the bacterium that decreases the most throughout the process (7 logarithmic units). With regard to *Enterococcus* sp., as in the previous case, a progressive decrease is observed during the process, achieving at the end a reduction of five logarithmic units in its concentration. As in the case of the aerated static pile, some increases in the concentration of the bacteria analyzed throughout the process may be due to the bacterial contribution of the irrigation water or to the non-homogeneity of the temperature in the pile. No significant differences in bacterial evolution are found in the cases of composting with an initial mixture of OFMSW and SM of 1:1 (*v*/*v*) or 1:2 (*v*/*v*), as was the case using the other two pilot facilities.

A comparison of the three facilities shows that the automatic reactor produces the highest total coliform removal, while the greatest reduction of *Escherichia coli* occurs in the turned pile. The latter may be due to the fact that *Escherichia coli* is a thermotolerant bacterium and it is in the turned pile where higher temperature values are reached throughout the composting process, although without exceeding the temperature of “microbial suicide”, conditions in which an inhibition of the biological treatment would occur. In the case of *Enterococcus* sp., the reductions obtained are similar in the three installations. Therefore, it cannot be stated that there is one type of facility among those used that produces a greater bacterial inactivation in general. It is important to have a good control of the temperature and to detect unfavorable points that could cause bacterial proliferation.

With regard to the fertilizing capacity of the compost obtained, in all cases high values of the germination index were obtained, proving that the compost is stable for use as an agricultural amendment (values above 84% in all cases for the calculation bases of distilled water and soil without input, and above 70% in all cases for the calculation base of commercial peat). According to Zucconi et al. (1981), germination rates below 50% show a strong presence of phytotoxic substances and, therefore, a compost not stable for use.

## 4. Conclusions

The composting process of OFMSW has been analyzed at pilot scale in three types of facility, automatic reactor, aerated static pile, and turned pile. All three performed correctly in terms of the monitoring of the physico-chemical parameters, resulting in a compost of sufficient stability and maturity to be applied on agricultural soil. Although the maximum temperature during the composting process was reached using the turned pile, this does not imply that this installation produces the greatest elimination of bacteria in general, but it does eliminate *Escherichia coli*, which is a thermotolerant bacterium. In all cases the bacterial concentrations in the final compost were lower than those detected in the initial components, complying with the requirements established in RD 506/2013 on fertilizer products (modified by RD 999/2017). However, it cannot be affirmed that there is a type of installation among those used that produces a greater bacterial inactivation in general. When composting with different types of facility, it is of interest to optimize the irrigation and aeration system to have a better control of the process and to study the possible temperature gradients in the piles to ensure good sanitization without the risk of bacterial proliferation posteriori. Finally, the different initial mixtures of OFMSW and SM used in this study did not have a significant influence on the functioning of the composting process or on the microbiological quality throughout the process. It should be pointed out that the irrigation water can provide a bacterial contribution that leads to increases in concentration during the composting process.

## Figures and Tables

**Figure 1 ijerph-19-15449-f001:**
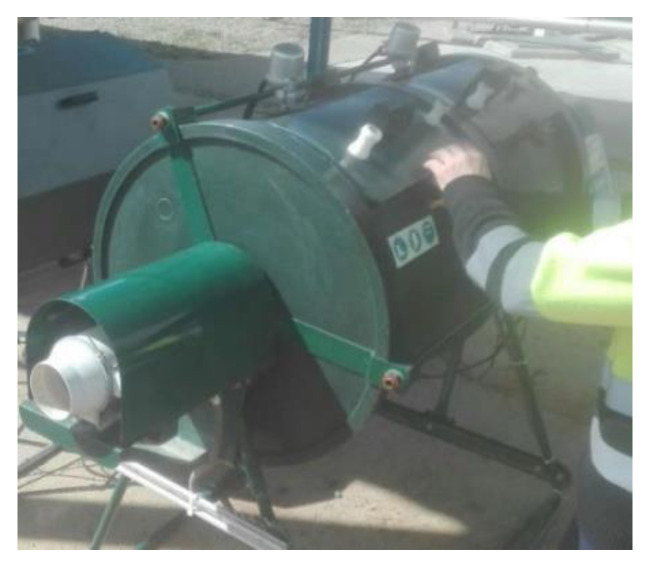
Automatic reactor.

**Figure 2 ijerph-19-15449-f002:**
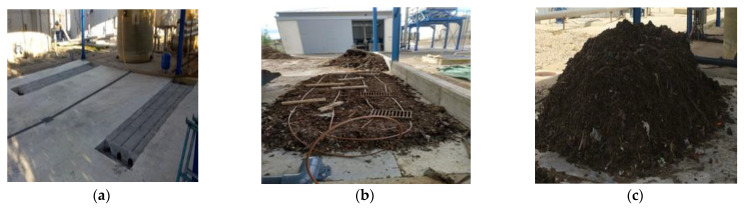
Installation of aerated static piles: (**a**) aeration system; (**b**) drip irrigation system; (**c**) aerated pile.

**Figure 3 ijerph-19-15449-f003:**
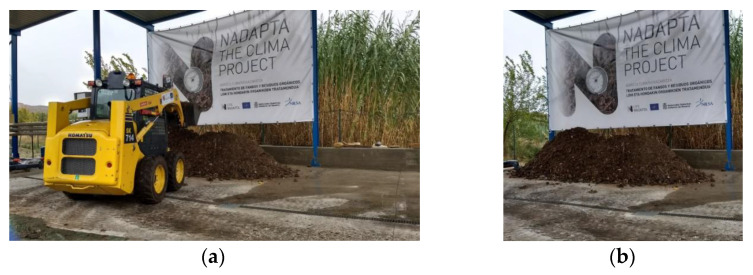
Preparation of turned piles: (**a**) mechanical turning; (**b**) turned pile.

**Figure 4 ijerph-19-15449-f004:**
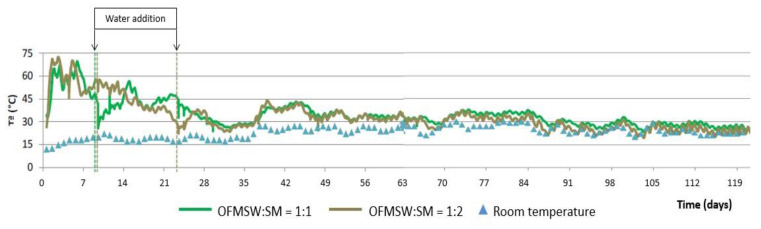
Temperature evolution in the composting process using an automatic reactor.

**Figure 5 ijerph-19-15449-f005:**
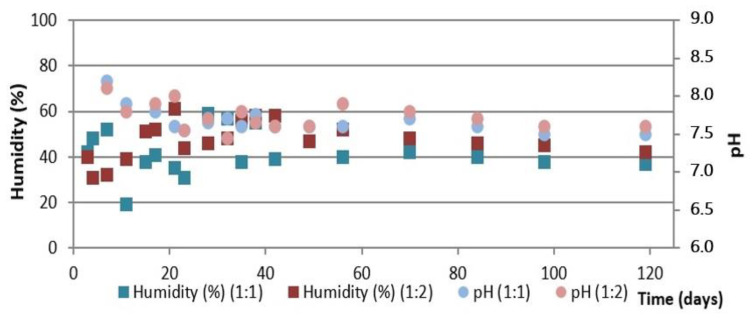
Evolution of humidity and pH in the composting process using an automatic reactor.

**Figure 6 ijerph-19-15449-f006:**
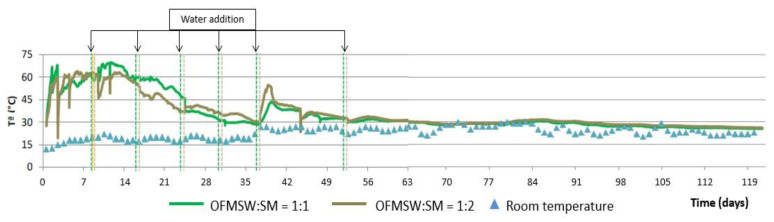
Temperature evolution in the aerated static pile composting process.

**Figure 7 ijerph-19-15449-f007:**
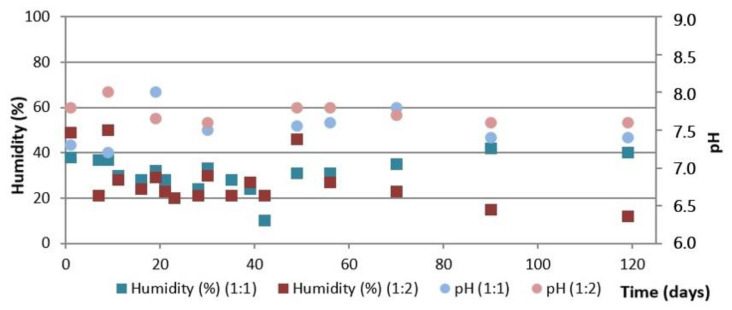
Evolution of humidity and pH in the aerated static pile composting process.

**Figure 8 ijerph-19-15449-f008:**
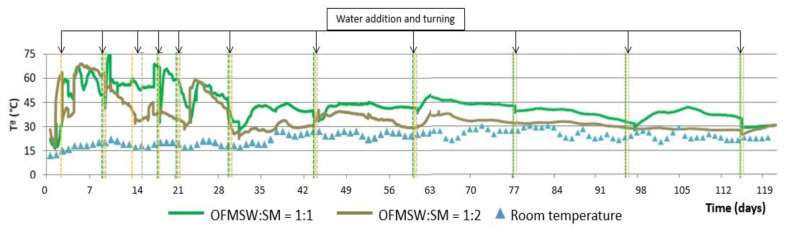
Temperature evolution in the turned pile composting process.

**Figure 9 ijerph-19-15449-f009:**
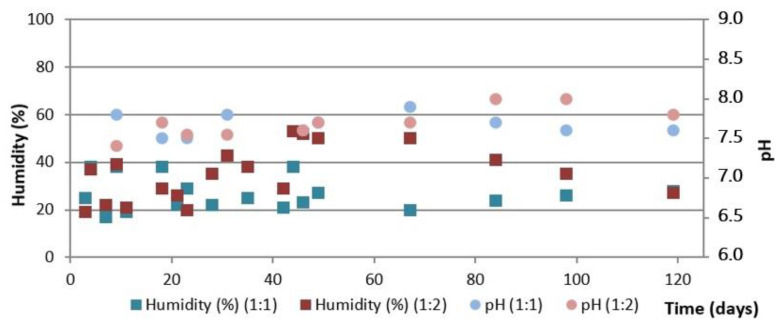
Evolution of humidity and pH in the turned pile composting process.

**Figure 10 ijerph-19-15449-f010:**
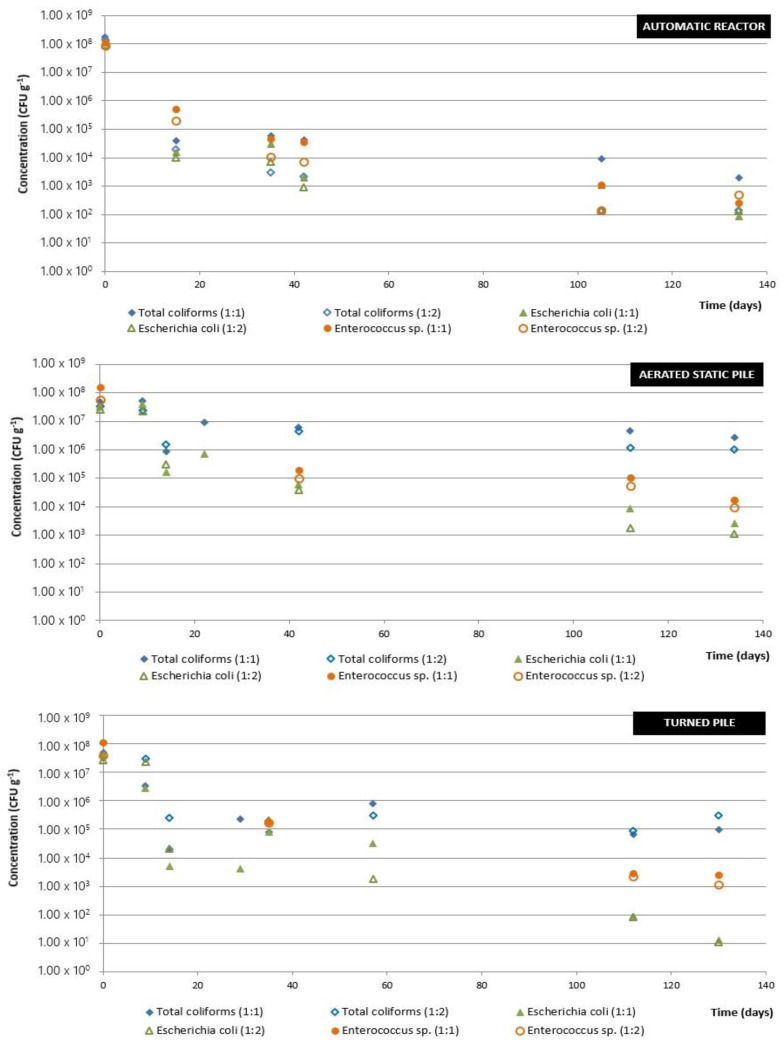
Evolution of bacterial concentrations during composting by the three systems used: automatic reactor, aerated static pile and turned pile, with ratios OFMSW/SM 1:1 or 1:2 (*v*/*v*).

**Table 1 ijerph-19-15449-t001:** Operational conditions of the composting processes during the study.

Type of Facility	Relation OFMSW: SM (*v*/*v*)	Initial Volume OFMSW (m^3^)	Initial Volume SM (m^3^)	Initial Mixing Volume (m^3^)	Starting Month	End Month
Automatic reactor	1:2	0.07	0.15	0.25	May	September
Aerated static pile		2.20	4.60	6.80		
Turned pile		2.20	4.60	6.80		
Automatic reactor	1:1	0.12	0.12	0.24		
Aerated static pile		3.40	3.40	6.80		
Turned pile		3.40	3.40	6.80		

**Table 2 ijerph-19-15449-t002:** Physico-chemical parameters, equipment, and standard methodology of analysis.

Parameter	Equipment	Standard Method	Reference
pH	Multiparametric meter Orion Star A3295	4500H^+^-B	[16]
Humidity (%)	Balance, heater	UNE-EN ISO 11465:2011	[17]
Total solids (%)	Balance, heater		
Volatile fraction (%)	Balance, muffle	2540G	[16]
Organic matter (% d.m.)	Carbon analyzer	5310B	
Nitrogen (% d.m.)	Kjeldahl equipment	4500-N	
Phosphorus (% d.m.)	Inductively Coupled Plasma Mass Spectrometer (ICP-MS)	4500-P	
Cadmium, copper, nickel, lead, zinc, mercury, chromium (mg kg^−1^)	Inductively Coupled Plasma and Optical Emission Spectrometer (ICP-OES)	3120B	

% d.m.: % on dry matter.

**Table 3 ijerph-19-15449-t003:** Microbiological parameters, culture media, and standard methods of analysis.

Bacteria	Culture Media	Standard Method	Reference
Total coliforms	Chromogenic Coliform Agar (CCA)	ISO 9308-1	[21]
		9215B-C-D	[16]
*Escherichia coli*	Chromogenic Coliform Agar (CCA)	ISO 9308-1	[21]
	Glucuronic Agar tryptone and bile (TBX)	9215B-C-D 9222D	[16]
*Enterococcus* sp.	Slanetx and Bartley Agar	ISO 7899-2	[22]
		9215B-C-D	[16]
*Clostridium perfringens*	SPS Agar	ISO 6461-2	[23]
*Salmonella* sp.	XLD Agar Chromogenic Agar Salmonella Latex test	ISO 6579-1	[24]
Total mesophiles	Nutritive Agar	9215B	[16]

**Table 4 ijerph-19-15449-t004:** Initial physico-chemical properties of OFMSW and SM.

Parameter	OFMSW	SM
pH	7.2 ± 0.1	7.8 ± 0.2
Humidity (%)	61.4 ± 2.5	31.6 ± 1.9
Total solids (%)	38.6 ± 1.4	68.4 ± 1.9
Volatile fraction (%)	76.4 ± 2.2	66.7 ± 2.1
Organic matter (% d.m.)	74.9 ± 0.9	72.6 ± 0.2
Nitrogen (% d.m.)	2.9 ± 0.1	1.2 ± 0.2
Phosphorus (% d.m.)	0.49 ± 0.05	0.24 ± 0.09
Cadmium (mg kg^−1^)	<0.4 ± 0.1	<0.4 ± 0.1
Copper (mg kg^−1^)	98 ± 2	50 ± 1
Nickel (mg kg^−1^)	28 ± 1	39 ± 1
Lead (mg kg^−1^)	29 ± 1	18 ± 1
Zinc (mg kg^−1^)	116 ± 3	108 ± 2
Mercury(mg kg^−1^)	<0.4 ± 0.1	<0.4 ± 0.1
Chromium (mg kg^−1^)	40 ± 2	59 ± 3

% d.m.: % on dry matter.

**Table 5 ijerph-19-15449-t005:** Initial microbiological properties of OFMSW and SM.

Bacteria	OFMSW (CFU g^−1^)	SM (CFU g^−1^)
Total coliforms	9.70 ± 5.30 × 10^7^	1.24 ± 1.23 × 10^7^
*Escherichia coli*	7.75 ± 5.25 × 10^7^	2.04 ± 1.10 × 10^4^
*Enterococcus* sp.	3.70 ± 2.60 × 10^8^	3.20 ± 0.70 × 10^3^
*Clostridium perfringens*	3.25 ± 1.75 × 10^4^	1.45 ± 0.25 × 10^2^
Total mesophiles	1.85 ± 1.16 × 10^9^	4.50 ± 0.70 × 10^7^
*Salmonella* sp.	Not detected	Not detected

## Data Availability

Not applicable.

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
