# Peer review of "Physico-Chemical and Microbiological Control of the Composting Process of the Organic Fraction of Municipal Solid Waste: A Pilot-Scale Experience"

_ijerph, 2022, doi:10.3390/ijerph192315449_

Round 1

Reviewer 1 Report (Previous Reviewer 2)

It is just fine.

This manuscript is a resubmission of an earlier submission. The following is a list of the peer review reports and author responses from that submission.

Round 1

Reviewer 1 Report

In general, all figures in section 3.2.1 should be changed. They are of poor quality, the axes are missing or incomplete. What do the dashed lines in figures 4, 6 and 8 mean? In my manuscript, in Figure 8 there are some blue lines under some arrows, what are they? They are not indicated anywhere.

In the text the figures and tables are named in a different way than in the figure or table caption (Example: Figure 4 is called figure 4).

When thermal values are given it is wrong to put 22ºC, the correct thing to do is to put ºC. Therefore, all the thermal values mentioned should be revised and the spacing should be corrected, as well as the symbology referring to the degrees.

Humidity should be around 50-60% to favor microbial activity. In the Turned pile the results in terms of humidity are confusing. In this type of pile, it is normal to humidify at the same time as turning.  This means that the humidity of the pile must be known beforehand in order to add the optimum volume of water. Throughout the process, 40% moisture is rarely reached. It is usually around 20-30 %. Why? How could this have affected the microbial results? 

In section 3.2.1. the x-axis shows up to 104 or 120 days or weeks (not specified in the figure) but in section 3.2.2. data are represented up to 140 days. Why?

RD 999/2017 is mentioned but neither the initial C/N ratio of the raw materials nor the one obtained at the end, which is one of the parameters indicated in legislation, is indicated. Why?

At the end of section 3.2.2. the phytotoxicity value is mentioned. On the other hand, why is a value of 84% optimal? Is this a phytostimulant? There is no discussion of the results it only indicates that it is optimal but not why.

They correlate temperature with the decrease in microbial load but do not mention if it complies with the EPA 2003 values or with the new values described in legislation. How many days were kept in thermophilic phase? Were they continuous or alternating?

In Figure 10, in the automatic reactor, a higher concentration of E.coli than of total coliforms is observed. It would be advisable to use the same color code for the three graphs in this figure.

In material and methods it is mentioned that Clostridium perfringes and total mesophilic microorganisms were analyzed, but the evolution of these results is not observed in the graphs. Why?

Response to reviewer 1:

Dear reviewer.

Below you can find the responses and changes made to the paper regarding your comments. Thank you for the review which we believe has helped to improve the paper.

1.- In general, all figures in section 3.2.1 should be changed. They are of poor quality, the axes are missing or incomplete. What do the dashed lines in figures 4, 6 and 8 mean? In my manuscript, in Figure 8 there are some blue lines under some arrows, what are they? They are not indicated anywhere.
In the sending and editing of the article there were some configuration problems in the formats of all the figures. This has been resolved.

2.- In the text the figures and tables are named in a different way than in the figure or table caption (Example: Figure 4 is called figure 4).
The naming of all tables and figures throughout the manuscript has been unified.

3.- When thermal values are given it is wrong to put 22ºC, the correct thing to do is to put ºC. Therefore, all the thermal values mentioned should be revised and the spacing should be corrected, as well as the symbology referring to the degrees.
This terminology has been corrected throughout the document.

4.- Humidity should be around 50-60% to favor microbial activity. In the Turned pile the results in terms of humidity are confusing. In this type of pile, it is normal to humidify at the same time as turning. This means that the humidity of the pile must be known beforehand in order to add the optimum volume of water. Throughout the process, 40% moisture is rarely reached. It is usually around 20-30 %. Why? How could this have affected the microbial results?
In fact, the moisture adjustment is carried out at the same time as the piles are turned to try to ensure that the water distribution is as homogeneous as possible. However, despite the fact that the piles were visually wet and adjusted, the analyses showed lower moisture content than expected. Even so, the adjustment is considered optimal because it is observed that in the first weeks after the addition of water there is an increase in temperature and a reduction in the microbiological concentration. We are currently working on optimising the irrigation and the method for determining the moisture content.

5.- In section 3.2.1. the x-axis shows up to 104 or 120 days or weeks (not specified in the figure) but in section 3.2.2. data are represented up to 140 days. Why?
The physico-chemical parameters were analysed for 120 days. Monitoring of these parameters is more continuous and they were not measured anymore once they were considered to be stabilised. In the case of the microbiological parameters, we decided to measure them over a longer period of time because, due to the complexity of the analyses, the monitoring is not so continuous and the previous values were not considered stable (in addition to not having bibliographical references to predict their behaviour beforehand).

6.- RD 999/2017 is mentioned but neither the initial C/N ratio of the raw materials nor the one obtained at the end, which is one of the parameters indicated in legislation, is indicated. Why?
In our study we start from components that are treated at industrial level so we are sure that they have an adequate C/N ratio. It is not the aim of this study to analyse the C/N ratios but we have some specific data (OFMSW, C/N=5.2-5.5; final compost, CN=8-13) that we have not included in the paper.

7.- At the end of section 3.2.2. the phytotoxicity value is mentioned. On the other hand, why is a value of 84% optimal? Is this a phytostimulant? There is no discussion of the results it only indicates that it is optimal but not why.
The germination rate is a % that expresses the germinated seeds and the growth attained by the radicle. It is a bioassay used to determine the stability of the compost. According to Zucconi et al. (1981), germination rates below 50% show a strong presence of phytotoxic substances and therefore a compost not stable for use; germination rates above 80% indicate no presence of phytotoxic substances and therefore a compost with sufficient maturity for use.
A paragraph to this effect has been added in the paper.

8.- They correlate temperature with the decrease in microbial load but do not mention if it complies with the EPA 2003 values or with the new values described in legislation. How many days were kept in thermophilic phase? Were they continuous or alternating?
P8_TA-PROV(2019)0306 (proposal for a Regulation of the European Parliament and of the Council laying down provisions relating to the placing on the market of fertiliser products bearing the CE marking) lays down time and temperature intervals for considering that there is a reduction of microbiological contamination. In particular, 65 °C or more for at least 5 days; 60 °C or more for at least 7 days; or 55 °C or more for at least 14 days. In the processes studied, these conditions are met as corroborated by continuously measured temperature and in-situ measurements in several areas of the pile.
A paragraph to this effect has been added in the paper.

9.- In Figure 10, in the automatic reactor, a higher concentration of E.coli than of total coliforms is observed. It would be advisable to use the same color code for the three graphs in this figure.
In the sending and editing of the article there were some configuration problems in the formats of all the figures. This has been resolved.

10.- In material and methods it is mentioned that Clostridium perfringes and total mesophilic microorganisms were analyzed, but the evolution of these results is not observed in the graphs. Why?
Initially, a more complete characterisation analysis was carried out to check the degree of microbiological contamination of the starting materials and the results of this analysis are shown in the paper. From among all the bacteria analysed, it was decided to choose 4 of them for two reasons: because they were bacteria included in specific legislation or in other legislation in which they were used as indicators of contamination; and because we were able to analyse them by our own means and given the sufficient time required for microbiological analyses to be carried out correctly. The analyses of clostridium perfringens and total mesophylls were outsourced.

Reviewer 2 Report

The manuscript is well organized and contains all the components you would expect however, no statistical tests were carried out among the pilot installations tested neither between the level of OFMSW/SM 1:1 and 1:2 v/v. The authors did a good job synthetizing the literature. Since the authors´aim was to study the influence on the physico-chemical and microbiological characteristics of the compost obtained by three types of installation used in the composting process at pilot scale (automatic reactor, aerated pile and turned pile) and of the characteristics and mixing ratio of the initial waste used (organic fraction of municipal solid waste and pruning material as structuring material) and no differences were found no answer to NILSA in strict sense could be given or in other words they did not come up with any of the three technologies tested to improve composting in Navarra. The rationale to use two mixtures of OFMSW/SM 1:1 and 1:2 v/v is not adequately explained. On the other hand, the C/N ratio of the six treatments was not calculated, in section 2.4.1, line 162 the authors mention the measurement of elemental carbon but is not shown in the results. It is hard to accept for the reader that the six treatments were not different.

Suggestion for the authors:

Do work on the C/N ratio of each treatment;

If possible determine electrical conductivity in the OFMSW and the SM;

Characterize the compost from the waste treatment plant in Navarra that provided the OFMSW for your experiments and compare, if feasible, this compost with the composts you obtained;

It would be helpful to the composting plant in Navarra to know how N2 is found at the end of the process perhaps if you still have your compost samples you could determine NH4-N and NO3-N concentrations.

Other questions:

The drip irrigation system in the aerated piles was at the bottom of the pile?

In line 196 you state that... "At the beggining of the microbiological analyses the bacterial concentration was unknown"... it is obvious isn´t it?

%s.m.s. stands for what?

How could you explain that total coliforms concentration is similar in both the OFMSW and the SM (line 228)?

In page 11, line 8, FORSU and ME should be changed by OFMSW and SM.

Conclusions is with capital C (page 19, line 1).

Do you have any concerns on N2O and CH4 generation during aerobic digestion?

Response to reviewer 2:

Dear reviewer,

Below you can find the responses and changes made to the paper regarding your comments. Thank you for the review which we believe has helped to improve the paper.

The manuscript is well organized and contains all the components you would expect however, no statistical tests were carried out among the pilot installations tested neither between the level of OFMSW/SM 1:1 and 1:2 v/v. The authors did a good job synthetizing the literature. Since the authors´aim was to study the influence on the physico-chemical and microbiological characteristics of the compost obtained by three types of installation used in the composting process at pilot scale (automatic reactor, aerated pile and turned pile) and of the characteristics and mixing ratio of the initial waste used (organic fraction of municipal solid waste and pruning material as structuring material) and no differences were found no answer to NILSA in strict sense could be given or in other words they did not come up with any of the three technologies tested to improve composting in Navarra. The rationale to use two mixtures of OFMSW/SM 1:1 and 1:2 v/v is not adequately explained. On the other hand, the C/N ratio of the six treatments was not calculated, in section 2.4.1, line 162 the authors mention the measurement of elemental carbon but is not shown in the results. It is hard to accept for the reader that the six treatments were not different.

  • - No statistical tests were carried out among the pilot installations tested neither between the level of OFMSW/SM 1:1 and 1:2 v/v: Regarding the statistical study of the results, we have tried to make an 1-factor ANOVA (for FMSW/SM ratio and type of installation factors). Prior to this analysis, the variance homogeneity test was performed (Levene test) but it did not yield results. For this reason, we do not continue with this parametric test. Due to this, a non-parametric test was tested (Kruskal-Wallis) and the results were not very consistent. We believe that to perform a correct statistical study, we should repeat the investigation for several years under the same conditions (we are working on it). That is really complicated due to the conditions in which the study has been done. For this reason, we present the results as a consequence of a pilot experience in real conditions.
  • - No answer to NILSA in strict sense could be given or in other words they did not come up with any of the three technologies tested to improve composting in Navarra: NILSA's objective was not really to find out which technology improved composting, but to find out whether these technologies produced different results in terms of microbiological monitoring of the process. As the results show that there are no major differences between one type of installation and conditions and others among those tested, NILSA was recommended to choose the installation on the basis of other criteria (such as ease of operation, maintenance, etc.). In fact, we are currently working on optimising the mixing, aeration and moisture supply process in the piles.
  • - The rationale to use two mixtures of OFMSW/SM 1:1 and 1:2 v/v is not adequately explained: Both ratios were chosen according to the available pruning material as long as the mixture with the organic residues provided an adequate carbon/nitrogen value. A paragraph to this effect has been added to the paper.
  • - The authors mention the measurement of elemental carbon but is not shown in the results: This was an error in the paper, it has been removed.

Suggestion for the authors:
1.- Do work on the C/N ratio of each treatment;
In our study we start from components that are treated at industrial level so we are sure that they have an adequate C/N ratio. It is not the aim of this study to analyse the C/N ratios but we have some specific data (OFMSW, C/N=5.2-5.5; final compost, CN=8-13) that we have not included in the paper.

2.- If possible determine electrical conductivity in the OFMSW and the SM;
We do not have conductivity data on the materials. We only have point values for OFMSW that are between 3 and 6 dS/m. This parameter was not selected to monitor the whole process but we will take it into account for future analysis.

3.- Characterize the compost from the waste treatment plant in Navarra that provided the OFMSW for your experiments and compare, if feasible, this compost with the composts you obtained;
The waste treatment plant in Navarra only provided us with the starting material. We did not have access to the results of the waste treatment they do there. In addition, the aim of the work is to monitor different parameters, mainly microbiological ones, given the scarce literature on the subject, for which we decided to carry out the study in pilot plants and not in a real facility to which we did not have access. In order to avoid confusion, the information related to the treatment carried out in this plant has been removed from the paper.

4.- It would be helpful to the composting plant in Navarra to know how N2 is found at the end of the process perhaps if you still have your compost samples you could determine NH4-N and NO3-N concentrations.
We do not have the samples at the moment so it is impossible for us to analyse these parameters. However, we will suggest to NILSA to incorporate these parameters in future studies.

Other questions:
5.- The drip irrigation system in the aerated piles was at the bottom of the pile?
No. The aeration system consists of channels at the bottom of the pile. This system is automated. The irrigation system is placed at different heights (at least 3) of the pile with drip tubes.

6.- In line 196 you state that... "At the beggining of the microbiological analyses the bacterial concentration was unknown"... it is obvious isn´t it? %s.m.s. stands for what?
Indeed, this sentence is redundant and has been removed from the paper. As for % s.m.s. is a language error, it has been appropriately changed to % d.m.: % on dry matter, and its meaning included at the foot of the table.

7.- How could you explain that total coliforms concentration is similar in both the OFMSW and the SM (line 228)?In reality, there is no concrete explanation either. In all the analyses we have carried out, the initial bacterial concentration was quite high in all the materials. This makes us think that we can use this microbiological parameter as an indicator of the evolution of the microbiological concentration, and that it can be used as an indicator, not only in urban wastewater, but also in other matrices, since it is initially present in high concentrations in them.

8.- In page 11, line 8, FORSU and ME should be changed by OFMSW and SM.
This has been changed.

9.- Conclusions is with capital C (page 19, line 1).
This has been changed.

10.- Do you have any concerns on N2O and CH4 generation during aerobic digestion?
We have some biogas meters that we use in other facilities, but for the moment, as it is an aerobic process and as we cannot carry out analyses with guarantees of correct measurement, they are not being carried out. However, we are considering how to carry out these analyses.